# Concentration of Selected Metalloproteinases and Osteocalcin in the Serum and Synovial Fluid of Obese Women with Advanced Knee Osteoarthritis

**DOI:** 10.3390/ijerph19063530

**Published:** 2022-03-16

**Authors:** Jaromir Jarecki, Teresa Małecka-Masalska, Ewa Kosior-Jarecka, Wojciech Widuchowski, Piotr Krasowski, Martina Gutbier, Maciej Dobrzyński, Tomasz Blicharski

**Affiliations:** 1Department of Rehabilitation and Orthopaedics, Medical University of Lublin, 20-059 Lublin, Poland; blicharski@vp.pl; 2Department of Orthopaedics and Traumatology, Regional Hospital in Chełm, 22-100 Chełm, Poland; piotr.krasowski@gumed.edu.pl; 3Physiology Department, Medical University of Lublin, 20-059 Lublin, Poland; tmalecka@gmail.com; 4Department Diagnostics and Microsurgery of Glaucoma, Medical University of Lublin, 20-079 Lublin, Poland; ekosior@poczta.onet.pl; 5Departament of Physiotherapy, The College of Physiotherapy, 50-038 Wrocław, Poland; sportmed@sportmed.com.pl; 6Department of Pediatric Dentistry and Preclinical Dentistry, Wrocław Medical University, Krakowska 26, 50-425 Wrocław, Poland; martina.gutbier@umw.edu.pl (M.G.); maciej.dobrzynski@umw.edu.pl (M.D.)

**Keywords:** osteoarthritis, knee, matrix metalloproteinases, osteocalcin, female, obesity

## Abstract

The aim of the study was to evaluate the levels of selected MMPs (matrix metalloproteinases) and osteocalcin in the serum and synovial fluid of obese women with osteoarthritis and their correlations with clinical status. The studied group consisted of 39 overweight females undergoing primary total knee arthroplasty due to osteoarthritis (OA). The staging of knee OA was evaluated according to the Ahlbӓck and Kellgren–Lawrence scores. Synovial fluid and peripheral blood samples were obtained. The levels of selected MMPs and osteocalcin were assessed using commercial ELISA kits. The mean value of MMP3 was significantly higher in patients with more advanced disease in both serum (*p* = 0.0067) and synovial fluid (*p* = 0.0328). The pro-MMP13 level tended to be higher in synovial fluid in the case of more advanced stages (*p* = 0.0882), with no tendency regarding the serum level (*p* = 0.9595). The synovial level of pro-MMP1 was significantly correlated with the synovial concentration of MMP9 and MMP3. The synovial level of MMP9 also showed a significant correlation with the synovial level of MMP3 and pro-MMP13. Furthermore, it was found that the serum level of MMP3 was significantly correlated with the synovial pro-MMP13 level. A correlation between the osteocalcin level in serum and its synovial level was determined. The serum MMP9 level showed a significant correlation with BMI, whereas the synovial MMP9 level was notably correlated with age. Our results showed that the levels of MMP3, MMP9, and pro-MMP13 increased in more advanced radiological stages of OA, indicating the underlying inflammatory process of OA.

## 1. Introduction

Osteoarthritis (OA), the most common type of arthritis, causes a progressive loss of joint function [1] and is characterized by biochemical, morphological, molecular, and biomechanical changes in both the extracellular matrix (ECM) and cells, which lead to cartilage softening, ulceration, fibrillation and loss, the sclerosis of subchondral bone as well as subchondral cysts, and osteophytes formation. OA develops as a combined result of environmental and genetic factors, where genetic factors account for nearly 50% of the risk [2].

The two main risk factors for knee OA are female gender and obesity. The incidence of OA has predilection to the female gender with a ratio of about 1.7:1 [3,4]. Women are not only more likely to have OA, but they also tend to have a more severe course [5]. Being obese or overweight have long been recognized as potent risk factors for OA, especially OA of the knee [6], as mechanical joint overload tends to be a main factor in the pathogenesis of OA.

The destruction of cartilage is a common pathological feature in OA and is the major cause of joint dysfunction, followed by the impairment of “quality of life”. Two pathways are known for the destruction of cartilage. First, an intrinsic pathway in which chondrocytes themselves degrade cartilage ECM, and second, an extrinsic pathway in which tissues or cells other than chondrocytes, such as inflamed synovium, OA pannus tissue, and infiltrated inflammatory cells, break down the cartilage ECM mostly through the synovial fluid (SF). In both pathways, the enzymatic digestion of the ECM is ascribed to cartilage destruction.

Matrix metalloproteinases (MMPs), a group of ECM-degrading enzymes expressed in joint tissues, are believed to play a crucial role in joint destruction during OA [7,8,9]. MMPs are a family of proteinases composed of at least 18 members, which are classified into five subgroups: collagenases, including tissue collagenase (MMP1) and collagenase 3 (MMP13); gelatinases, such as gelatinase B (MMP9); stromelysins, including stromelysin 1 (MMP3); membrane-type MMPs (MT-MMPs); and other MMPs. The enzymatic activities of these MMPs are strictly controlled by inhibition with specific inhibitors, the tissue inhibitors of metalloproteinases (TIMPs). Therefore, the balance between the levels of MMPs and TIMPs in the SF and local tissues may be a determinant of the possibility of MMPs to attack the cartilage ECM [10].

Different studies have demonstrated increased subchondral bone turnover accompanied by specific architectural changes in the subchondral trabecular bone in OA joints. The functional integrity of the articular cartilage can therefore depend on the mechanical properties of the underlying bone [11]. Osteocalcin (OC) plays a key role in both the biological and mechanical functions of bone. It is the most abundant non-collagenous protein (NCP) in bone and is specifically expressed in osteoblasts [12]. It is produced during bone formation, late in the mineralization process, and controls—directly and/or indirectly—bone mass, mineral size, and orientation [13]. Further, OC is involved in organizing the ECM. “Osteocalcin has been thought as a serum marker of bone formation and mineralization”; however, new evidence now points to an additional hormonal role. These newly described properties link the energy demands of bone to endocrine pathways associated with the nutrient availability, leptin, adiponectin, and insulin within the skeletal metabolism [14,15,16,17,18]. Additionally, there are some studies combining the action of osteocalcin and MMPs in different diseases [19,20].

The aim of this study was to evaluate the level of selected MMPs and OC in the serum and SF of obese women with OA and to look for correlations with their clinical status.

## 2. Materials and Methods

The studied group constituted of 39 female patients undergoing primary total knee arthroplasty (TKA) because of knee OA at the Department of Orthopaedics and Traumatology of Locomotor System in Regional Hospital in Chelm, Poland in 2019. Every patient who met the inclusion criteria, did not meet any of the exclusion criteria, and signed a consent agreement form was included to the study group. The study followed the guidelines of the Declaration of Helsinki and the project was accepted by the Local Ethical Committee at Medical University of Lublin, Poland (KE-0254/39/2019).

The inclusion criteria of the studied group were as follows: female gender, age over 50 years, advanced primary knee OA, and BMI over 25. Each patient’s height and weight were measured in order to calculate their body mass index according to the formula BMI = weight [kg]: height [m]^2^.

The exclusion criteria for this study were as follows:Male gender;Age under 50;Previous surgical history on the knee prepared for TKA;Previous injections to the knee with planned surgery with steroids or hyaluronans;History of knee trauma;Ongoing knee infections or infection within 3 months before the planned surgery;Ongoing or previous history of deep vein phlebitis;Active dental or periodontal diseases;Intake of any symptomatic slow acting drugs for OA.

On the day before the surgery, X-rays were performed on both knees of all patients in a standing position, with both anterior-posterior and lateral views. The OA stage was evaluated according to the clinical Knee Society Score (KSS) scale and modified radiological scales according to Ahlbäck [21] and to Kellgren–Lawrence [22]. All assessments were performed by two orthopedic surgery specialists (J.J. and T.B.). The presented results are mean values.

The Kellgren–Lawrence (K-L) system is a common method for classifying the severity of knee OA using five grades as follows:Grade 0: no radiographic features of OA are present.Grade 1: doubtful joint space narrowing (JSN) and possible osteophytic lipping.Grade 2: definite osteophytes and possible JSN on anteroposterior weight-bearing radiograph.Grade 3: multiple osteophytes, definite JSN, sclerosis, and possible bony deformity.Grade 4: large osteophytes, marked JSN, severe sclerosis, and definite bony deformity.

The other applied classification was proposed by Ahlbäck et al. in 1968. According to the Ahlbäck system, knee joint osteoarthritis can be classified as:Grade 1: joint space narrowing (less than 3 mm).Grade 2: joint space obliteration.Grade 3: minor bone attrition (0–5 mm).Grade 4: moderate bone attrition (5–10 mm).Grade 5: severe bone attrition (more than 10 mm).

The day before the surgery, peripheral blood samples from all patients were obtained and centrifuged. The serum was frozen and stored at −40 °C until further tests. Before the surgery started, a tourniquet was placed at the proximal thigh. During TKA, before the incision of the knee joint capsule, the SF was obtained from the suprapatellar recess with a 12 G needle, then portioned, frozen and stored at −40 °C until further studies. In the case of contamination of the SF with blood, the patient was excluded from the study. Tranexamic acid was not used during the surgery.

The SF and serum were defrosted before the tests. For the evaluation of the concentrations of pro-MMP1, MMP3, MMP9, pro-MMP13, and OC, the following commercial ELISA kits were used: human pro-MMP-1 Quantikine ELISA Kit (R&D System DMP100), human MMP-9 Quantikine ELISA Kit (R&D System DMP900), human total MMP-3 Quantikine ELISA Kit (R&D System DMP300), human pro-MMP-13 Quantikine ELISA Kit (R&D System DM1300), and human osteocalcin Quantikine ELISA Kit (R&D System DSTCN0). All tests were conducted according to the manufacturers’ instructions at the Chair and Department of Human Physiology, Medical University of Lublin, Poland.

A statistical analysis of the results was performed with the Statistica 13 software, and *p* < 0.05 was considered statistically significant. The results were reported mainly as a mean ± SD or percentage values. Normal distribution was checked with the Kołmogorow–Smirnow test. In the case of continuous variables, a Pearson test was used to compare normally distributed data and a Spearman test was used in the case of non-normally distributed data. In the case of non-normal distribution of all studied markers, the non-parametric U-Mann–Whitney test was used.

## 3. Results

The demographic and clinical characteristics of the studied group are shown in Table 1. The comparison of the MMPs and osteocalcin levels were also performed for clinical and functional KSS knee scores, as shown in Table 2. After dividing the studied group according to the stage of disease in the K-L scale, no differences in the studied parameters were noted. When the staging was performed using the Ahlbӓck scoring, the mean value of MMP3 was significantly higher in patients with more advanced disease for both serum (*p* = 0.01) and SF (*p* = 0.03). Additionally, pro-MMP13 tended to be higher in the SF of patients with more advanced stages evaluated according to the Ahlbӓck (*p* = 0.09) and K-L (*p* = 0.14) grading systems, whereas in no difference could be observed in the serum level (Ahlbӓck score (*p* = 0.95) and K-L score (*p* = 0.91)). In case of SF MMP9 level, there was a statistical tendency for a higher concentration in more advanced stages assessed according to the Ahlbӓck scale (*p* = 0.06), also without differences in the serum level (*p* = 0.99). The increase in serum MMP3, SF MMP3, and SF pro-MMP13 with the progression of OA was also visible when the patients were divided according to single grades in the Ahlbӓck scale, although without statistical significance (Table 3). The details of the correlations observed for the studied factors regarding the Ahlbӓck scale are shown in Table 4, and those concerning the K-L grading system in Table 5. In Table 6, the observed results of relations between the studied factors are presented. The SF level of pro-MMP1 was significantly correlated with the SF concentration of MMP9 (R = 0.53, moderate strength) and MMP3 (R = 0.49, moderate strength). The SF level of MMP9 was significantly correlated with the SF level of MMP3 (R = 0.44, moderate strength) and pro-MMP13 (*p* = 0.35, weak strength). Additionally, the serum level of MMP3 was significantly correlated with the SF pro-MMP13 level (R = 0.66, moderate strength). The OC level in serum was correlated with its SF level (R = 0.65, moderate strength).

Additionally, the correlations of the studied factors with demographic features were examined (Table 7). Serum MMP9 level was significantly correlated with BMI (R = 0.40, weak strength), whereas SF MMP9 level was significantly correlated with age (R = 0.33; weak correlation).

## 4. Discussion

OA is typically described as a heterogeneous disease with a wide range of possible underlying pathways, which lead to a similar outcome—joint destruction. Female gender and obesity are among the most important risk factors of knee OA, which is why this study focused on patients with a combination of both. Elevated BMI influences the risk of OA by some possible mechanisms. Obesity not only increases mechanical stress on the tibiofemoral cartilage, but also causes intra-articular inflammation. Compressive stress induces the activation of mechanoreceptors at the chondrocyte surface [23], resulting in a catabolic phenotype characterized by enhanced MMP production. Adipose tissue has a pivotal role, being the additional source of cytokines, chemokines, and adipokines [24].

MMP1 is produced by synovial cells, chondrocytes, and osteoblasts. It is an interstitial collagenase capable of degrading interstitial collagens (types I, II, and III) and is thought to be a multifunctional molecule with important roles in diverse physiologic processes [25]. In pathological conditions, MMP1 expression increases, resulting in excessive connective tissue destruction [26]. During OA, MMP1 is expressed at a high level in chondrocytes, which suggests its crucial role in OA pathogenesis. In some studies, the concentration of MMP1 in the SF and serum were similar when comparing OA patients and healthy individuals [27,28], whereas a pronounced elevation was detected in other studies [29,30]. In this study, the pro-MMP1 level was higher in SF than in serum. Anitua et al. assessed the MMP1 levels in OA patients and found that the plasma and SF results did not correlate, similar to our results. In fact, the serum level of MMPs is influenced by food intake and circadian and activity-related variations [31], which were not taken into consideration in this study.

Although MMP1 is crucial during OA pathogenesis, we could not find any significant correlations between the pro-MMP1 level and clinical staging in both serum and SF. This may be connected with the fact that pro-MMP1 may not reflect the level of active MMP1 [32]. Some researchers [33,34] have reported relatively high MMP1 level in SF at the early OA stage, and a decrease in levels with the progression of OA. In our results, when the evaluation of the staging was performed using the Kellgren–Lawrence system, decreasing pro-MMP1 levels in SF were also observed with OA progression; however, there was no major significance. The decline in pro-MMP1 levels in SF with the progression of OA may be related to the deterioration of superficial cartilage layers during more advanced OA. Moreover, some studies have shown a negative correlation between MMP1 expressions and both age and BMI [35], which was not confirmed in this study.

MMP3 (stromelysin 1) has been thought to play an essential role in the degradation of the cartilage matrix [36]. It is produced by synovial fibroblasts [37] and by chondrocytes in the articular cartilage during the course of OA [38]. After activation, MMP3 is capable of degrading many cartilage ECM components and can also activate other MMPs, including pro-MMP1, pro-MMP7, pro-MMP8, pro-MMP9, and pro-MMP13 [39]. Increased serum levels of MMP3 have been observed in rheumatoid arthritis (RA) as well as OA [40,41,42]. In this study, the MMP3 level in SF was more than 10 times higher compared to the serum level, which may point to the massive destructive process within the knee cartilage. Masuhara et al. found that the mean serum MMP3 level in women with rapidly destructive hip OA was 2–3 times higher than that in women with typical hip OA [43]. MMP3 has an inhibitory role in hyperplastic adipose expansion; its expression declines markedly within 3 days of the addition of adipogenic compounds [44], which may be the reason for the low concentration of MMP3 in the serum of obese females observed in this study.

Yoshihara et al. [42] found that MMP3 levels were enhanced in the early and middle stages of RA and decreased in the advanced stage. Thus, these data, in addition to the findings from MMP3 deficient mice, suggest that MMP3 may have a crucial role in the initiation and progression of cartilage destruction. In our study, the SF level of MMP3 was high and similar in advanced OA, though the earliest OA stages were not examined and compared. Additionally, MMP3 levels, both in serum and SF, correlated with the radiological Ahlbӓck score, showing an increase in MMP3 concentrations in more advanced clinical stages, which was also observed in other studies [45,46,47]. This indicates a systemic inflammatory state in knee OA patients. The active MMP3 is able to activate the procollagenases (pro-MMPs1, −8, and −13) and pro-MMP9 [48]; this is supported by the correlations between the SF level of MMP3 and pro-MMP1 and MMP9 observed in this study.

MMP9 is the most complex member of the MMP family with its highly destructive proteolytic activity against the many types of collagen that form cartilage [49]. Additionally, MMP9 has been reported to be a marker of obesity [46,50]. In this study including obese females, a correlation between serum MMP9 and BMI was observed. MMP9 protein expression levels have been reported to be either correlated with OA or not related at all [51]. In one meta-analysis [52], the high protein level of MMP9 proteins in OA was found to accelerate the pathogenesis of the disease. Similarly, in this study, MMP9 levels tended to increase in SF in more advanced knee OA stages and were higher in serum than in SF. A higher level of MMP9 has been detected in the SF of RA and inflammatory arthritis compared with the SF in OA [53].

Increased collagenase-3 (MMP13) activity plays an important role in the induction and pathogenesis of OA [54] by causing articular cartilage degradation and pathological changes in joints [55]. MMP13 is particularly expressed in the OA cartilage but not in healthy patients [56]. However, recent studies have revealed that although human chondrocytes isolated from healthy adults constitutively express and secrete MMP13, it is rapidly degraded by chondrocytes [57]. MMP13 is highly active against collagen type II, the predominant collagen in cartilage [58]. Pallu et al. showed the BMI-dependent effect of leptin for the expression of TIMP2 and MMP13 [59]. In this study, pro-MMP13 tended to be higher in the SF of patients with more advanced stages, whereas there was no difference in serum level. On the contrary, Sato [60] showed that MMP13 was abnormally expressed during different OA stages and was upregulated during the early stage and downregulated during the late stage in human OA cartilage. Additionally, studies have shown that mechanical stress upregulates MMP13 expression rapidly in chondrocytes in vitro [61].

In this study, the SF levels of MMP9 were significantly correlated with the synovial levels of pro-MMP13, which confirms the results of previous studies [62]. This finding suggests that MMP9 and pro-MMP13 in arthritic SF might be involved in joint degradation. MMP9, which inhibits both angiogenesis in the pannus and the activation of pro-MMP13 in SF, may play a pivotal role in the development of a MMP inhibitor and therapeutic drug for arthritis [62]. Moreover, leptin, a possible key mediator linking obesity to OA, modulates the degradative functions of the chondrocytes through the upregulation of MMP9 and −13 [63].

OC has been proposed as a biomarker for OA detection and monitoring. Its biologic function is considered to be related to the inhibition of bone matrix mineralization [64]. Dieppe et al. [65] observed that increased bone remodeling correlated with the severity of OA, and increased OC production in OA osteoblasts has been reported [66]. Berry et al. showed that higher levels of OC formation were associated with reduced cartilage loss [67]. Salisbury et al. [68] observed that in OA, the levels of SF OC were significantly lower than the serum levels and that SF OC concentrations were directly correlated with serum concentrations, which is in accordance with our results. Additionally, in our study, the correlation between OC levels in serum and SF and OA stage were not found, which has been observed previously [69,70]. Bruyere et al. found that the 3-year radiological progression of knee OA could be predicted by a 1-year increase in OC [71,72]. Van Spil et al. found positive associations between bone degradation and synthesis markers (OC) with the presence of early-stage radiographic knee OA, which may indicate that a high bone turnover state is an aggravating factor in the early stages of radiographic knee OA.

In our study, a negative correlation between serum OC level and SF level of MMP9 was observed, which reflects the contradictory actions of these two OA biomarkers. The positive correlation between pro-MMP13 and OC levels in serum may be connected with their downregulation in advanced OA stages. Varga et al. showed an interrelationship between OC and MMP13 expression: externally added OC attenuated the MMP13 expression dose dependently and, furthermore, increased vitamin D-stimulated MMP13 expression, which suggests that OC may be a modulator of hormonally regulated MMP13 expression [73].

OC acts as a hormone by stimulating insulin production and sensitivity in body organs. Animal studies have shown that an increase in serum OC concentration prevents obesity and glucose intolerance [74]. Previous research has described OC levels to be inversely associated with BMI and positively associated with age and female gender [73,75], which was not confirmed in our study. We involved only the patients with advanced OA and obesity, and the OC levels may reflect the enhanced local pathological process hiding a possible relation to general risk factors.

This study has some limitations. The results are based on patients from one center, and the population study could be more conclusive. All patients undergoing TKA have an advanced stage of the disease. However, the material from earlier phases could only be obtained during arthroscopy, which is a different surgical procedure; additionally, obtaining samples from a healthy control group seems to be unethical. Finally, the authors decided to focus on the pro-forms of some MMPs, which reflect their production but are not always strictly related to their active forms [32].

## 5. Conclusions

Our results showed that the levels of MMP3, MMP9, and pro-MMP13 increased with a more advanced radiological grade of OA, indicating an underlying inflammatory process in OA. A better understanding of the mechanisms underlying their cooperative pathways during the different stages of knee OA may help build a space to search for improved prophylaxis and treatment in the future.

## Figures and Tables

**Table 1 ijerph-19-03530-t001:** Demographic and clinical characteristics of the studied group.

Variable	Mean Value +/− SD or *n* (%)	Minimum	Maximum
Number of Patients	39		
Age (years)	69.10 +/− 6.46	52	85
Body Mass Index (kg/m^2^)	31.52 +/− 5.55	25.48	46.90
Duration of Symptoms (years)	7.13 +/− 6.38	2	40
Ahlbӓck Score	3.05 +/− 1.10	1	5
Kellgren–Lawrence Score	3.69 +/− 0.57	2	4

**Table 2 ijerph-19-03530-t002:** The correlation between the level of studied MMPs, osteocalcin, and KSS clinical and functional scales.

Variable	KSS Clinical	KSS Functional
pro-MMP1 serum [ng/mL]	0.06	−0.21
pro-MMP1 synovial fluid [ng/mL]	−0.07	−0.18
MMP3 serum [ng/mL]	−0.23	−0.22
MMP3 synovial fluid [ng/mL]	−0.22	−0.31 *
MMP9 serum [ng/mL]	0.03	−0.27
MMP9 synovial fluid [ng/mL]	−0.19	−0.19
MMP13 serum [ng/mL]	0.04	−0.24
MMP13 synovial fluid [ng/mL]	−0.37 *	−0.25
Osteocalcin serum [ng/mL]	−0.4 *	0.03
Osteocalcin synovial fluid [ng/mL]	−0.17	0.06

Table shows R values; *—statistical significance.

**Table 3 ijerph-19-03530-t003:** Levels of studied MMPs and osteocalcin in different stages of OA.

Variable	All	Ahlbäck I–III	Ahlbäck IV–V	*p*	K-L I–III	K-L IV	*p*
		Mean +/− SD	Mean +/− SD		Mean +/− SD	Mean +/− SD	
*n*	39	27	12		10	29	
pro-MMP1 serum [ng/mL]	0.66 +/− 1.00	0.77 +/− 1.15	0.38 +/− 0.46	0.26	0.92 +/− 1.81	0.57 +/− 0.55	0.34
pro-MMP1 SF [ng/mL]	17.90 +/− 9.99	17.18 +/− 10.31	19.58 +/− 9.42	0.49	19.42 +/− 9.79	17.40 +/− 10.17	0.58
MMP9 serum [ng/mL]	100.26 +/− 84.36	100.40 +/− 78.26	99.95 +/− 101.00	0.99	93.12 +/− 57.62	102.65 +/− 92.29	0.76
MMP9 SF [ng/mL]	41.64 +/− 56.96	30.66 +/− 41.55	67.25 +/− 79.01	0.06	38.71 +/− 58.43	42.62 +/− 57.44	0.85
MMP3 serum [ng/mL]	7.02 +/− 5.49	5.52 +/− 2.62	10.50 +/− 8.42	0.01 *	5.81 +/− 2.11	7.42 +/− 6.20	0.43
MMP3 SF [ng/mL]	174.97 +/− 37.83	166.70 +/− 37.55	194.26 +/− 32.13	0.03 *	177.78 +/− 31.22	174.03 +/− 40.23	0.79
pro-MMP13 serum [pg/mL]	32.12 +/− 32.32	32.29 +/− 35.09	31.71 +/− 26.08	0.95	33.16 +/− 28.38	31.77 +/− 33.97	0.91
pro-MMP13 SF [pg/mL]	291.18 +/− 359.66	227.70 +/− 303.72	439.28 +/− 444.99	0.09	146.90 +/− 257.07	339.27 +/− 379.31	0.14
Osteocalcin serum [ng/mL]	30.28 +/− 38.89	33.27 +/− 44.59	23.29 +/− 20.14	0.46	17.23 +/− 14.73	34.62 +/− 43.46	0.22
Osteocalcin SF [ng/mL]	10.969 +/− 14.226	12.2816 +/− 16.8075	7.90 +/− 3.09	0.38	8.97 +/− 4.11	11.63 +/− 16.28	0.61

*—statistically significant.

**Table 4 ijerph-19-03530-t004:** Levels of studied MMPs and osteocalcin and epidemiologic risk factors in different stages of OA according to the Ahlbäck scale.

Variable	1	2	3	4	5	ANOVA	*p*
	Mean +/− SD	Mean +/− SD	Mean +/− SD	Mean +/− SD	Mean +/− SD	F	
*n*	4	6	17	8	4		
pro-MMP1 serum [ng/mL]	0.27 +/− 0.28	1.36 +/− 2.30	0.69 +/− 0.58	0.43 +/− 0.55	0.29 +/− 0.25	0.87	0.42
pro-MMP1 SF [ng/mL]	13.67 +/− 9.86	23.26 +/− 8.38	15.94 +/− 10.65	18.11 +/− 10.33	22.54 +/− 7.65	1.53	0.23
MMP9 serum [ng/mL]	45.08 +/−31.24	125.15 +/− 48.10	104.45 +/− 89.00	100.66 +/− 119.66	98.53 +/−63.18	0.49	0.62
MMP9 SF [ng/mL]	15.86 +/− 14.81	53.95 +/− 72.92	26.19 +/− 29.58	59.18 +/− 89.03	83.41 +/− 62.23	0.24	0.79
MMP3 serum [ng/mL]	7.154 +/−1.97	4.92 +/− 1.82	5.36 +/− 2.90	10.29 +/− 9.82	10.94 +/− 5.89	0.66	0.52
MMP3 SF [ng/mL]	158.73 +/− 32.65	190.47 +/− 25.13	160.54 +/− 40.14	191.35 +/− 33.39	200.10 +/− 33.41	0.05	0.95
pro-MMP13 serum [pg/mL]	47.05 +/− 42.91	23.90 +/− 9.36	31.81 +/− 39.10	31.05 +/− 11.23	33.05 +/− 46.87	0.27	0.77
pro-MMP13 SF [pg/mL]	99.85 +/− 45.11	178.26 +/− 338.79	272.60 +/− 324.88	360.62 +/− 358.55	596.60 +/− 613.66	1.08	0.35
Osteocalcin serum [ng/mL]	13.87 +/− 3.40	19.47 +/− 19.20	42.18 +/− 52.99	21.72 +/− 21.70	26.44 +/− 19.19	0.75	0.48
Osteocalcin SF [ng/mL]	8.89 +/−4.39	9.02 +/−4.33	14.12 +/−120.73	7.94 +/−3.65	7.84 +/−1.98	0.15	0.87
BMI	33.68 +/− 2.86	31.46 +/− 4.75	31.66 +/− 5.61	31.45 +/− 6.35	29.78 +/− 8.00	0.24	0.92
Age	65.5 +/− 13.2	69.5 +/− 4.1	69.5 +/− 6.4	70.5 +/− 5.8	69.0 +/− 2.1	0.40	0.81

**Table 5 ijerph-19-03530-t005:** Levels of studied MMPs and osteocalcin and epidemiologic risk factors in different stages of OA according to the Kellgren–Lawrence scale.

Variable	2	3	4	ANOVA	*p*	Post Hoc NIR
	Mean +/− SD	Mean +/− SD	Mean +/− SD	F		
*n*	2	8	29			
pro-MMP1 serum [ng/mL]	0.35 +/− 0.43	1.07 +/− 2.02	0.57 +/− 0.55	0.87	0.42	
pro-MMP1 SF [ng/mL]	9.09 +/− 7.48	22.01 +/− 8.79	17.40 +/− 10.17	1.53	0.23	
MMP9 serum [ng/mL]	42.17 +/− 29.10	105.86 +/− 56.75	102.65 +/− 92.29	0.49	0.62	
MMP9 SF [ng/mL]	14.27 +/− 6.76	44.83 +/− 64.57	42.62 +/− 57.44	0.24	0.79	
MMP3 serum [ng/mL]	8.73 +/− 1.20	5.086 +/− 1.59	7.42 +/− 6.20	0.66	0.52	
MMP3 SF [ng/mL]	173.44 +/− 0.72	178.86 +/− 35.31	174.03 +/− 40.23	0.05	0.95	
pro-MMP13 serum [pg/mL]	18.10 +/− 22.91	36.92 +/− 29.66	31.77 +/− 33.97	0.27	0.77	
pro-MMP13 SF [pg/mL]	128.30 +/− 5.09	151.55 +/− 291.27	339.27 +/− 379.31	1.08	0.35	
Osteocalcin serum [ng/mL]	12.96 +/− 5.50	18.30 +/− 16.38	34.62 +/− 43.46	0.76	0.48	
Osteocalcin SF [ng/mL]	7.17 +/− 0.76	9.42 +/− 4.52	11.63 +/− 16.28	0.15	0.87	
BMI	31.36 +/− 0.38	32.60 +/− 4.58	31.35 +/− 5.94	0.16	0.86	
Age	76.5 +/− 4.9	67.7 +/− 7.9	69.7 +/− 5.7	2.75	0.08	2 vs. 3 *p* = 0.0338

**Table 6 ijerph-19-03530-t006:** Correlations between studied MMPs and osteocalcin in serum and SF.

Variable	pro-MMP1 Serum [ng/mL]	pro-MMP1 SF [ng/mL]	MMP9 Serum [ng/mL]	MMP9 SF [ng/mL]	MMP3 Serum [ng/mL]	MMP3 SF [ng/mL]	pro-MMP13 Serum [pg/mL]	pro-MMP13 SF [pg/mL]	Osteocalcin Serum [ng/mL]	Osteocalcin SF [ng/mL]
pro-MMP1 serum [ng/mL]	X	0.056	0.11	−0.08	0.03	−0.02	0.04	0.02	−0.08	−0.14
pro-MMP1 SF [ng/mL]	0.056	X	−0.04	0.54 *	0.27	0.49 *	0.24	0.24	−0.01	−0.08
MMP9 serum [ng/mL]	0.11	−0.04	X	0.05	−0.21	0.11	0.27	−0.03	−0.15	−0.07
MMP9 SF [ng/mL]	−0.08	0.54 *	0.05	X	0.28	0.44 *	0.07	0.35 *	−0.06	−0.13
MMP3 serum [ng/mL]	0.03	0.27	−0.21	0.28	X	0.21	−0.21	0.66 *	0.01	0.03
MMP3 SF [ng/mL]	−0.02	0.49 *	0.11	0.44 *	0.21	X	0.05	0.19	−0.23	−0.02
pro-MMP13 serum [pg/mL]	0.04	0.24	0.27	0.07	−0.21	0.05	X	−0.24	0.02	0.0003
pro-MMP13 SF [pg/mL]	0.02	0.24	−0.03	0.35 *	0.66 *	0.19	−0.24	X	0.07	−0.09
Osteocalcin serum [ng/mL]	−0.08	−0.01	−0.15	−0.06	0.01	−0.23	0.02	0.07	X	0.65 *
Osteocalcin SF [ng/mL]	−0.14	−0.08	−0.07	−0.13	0.03	−0.02	0.0003	−0.09	0.65 *	X

Table shows R values; *—statistical significance.

**Table 7 ijerph-19-03530-t007:** Correlations between level of studied MMPs and osteocalcin, and epidemiologic risk factors.

Variable	Obesity	Age	BMI
pro-MMP1 serum [ng/mL]	0.84	−0.05	−0.23
pro-MMP1 SF [ng/mL]	0.70	0.05	−0.19
MMP9 serum [ng/mL]	0.25	−0.08	0.40 *
MMP9 SF [ng/mL]	0.49	0.33 *	−0.15
MMP3 serum [ng/mL]	0.84	0.22	−0.16
MMP3 SF [ng/mL]	0.52	0.23	−0.09
pro-MMP13 serum [pg/mL]	0.53	−0.24	−0.05
pro-MMP13 SF [pg/mL]	0.46	0.12	0.10
Osteocalcin serum [ng/mL]	0.75	0.10	−0.05
Osteocalcin SF [ng/mL]	0.65	0.23	−0.13

Table shows R values; *—statistical significance.

## Data Availability

The data presented in this study are available on request from the corresponding author.

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
