# Peer review of "Concentration of Selected Metalloproteinases and Osteocalcin in the Serum and Synovial Fluid of Obese Women with Advanced Knee Osteoarthritis"

_ijerph, 2022, doi:10.3390/ijerph19063530_

Round 1

Reviewer 1 Report

This is an observational clinical study with the description of matrix metalloproteinase levels in a cohort of women with osteoarthritis. The association between MMPs and osteocalcin is not new. The authors have done a some literature research and incorporated this in their work. A number of issues, however, need to be addressed and these mainly relate to the limitations of this type of studies in general. For instance, it would have been interesting to provide comparisons with the levels in other disease entities or provide links why osteocalcin was analysed aside multiple MMPs.

Major issues

  1. Introduction: it might have been interesting to provide information whether osteocalcin has been evaluated as a substrate of any MMP. In rather old studies (e.g. from the labs of Martel-Pelletier and of Tuckerman) the association between MMPs and osteocalcin was already made. A simple experiment proving that osteocalcin is a substrate of any MMP, which is found to be increased in the present and other studies, would have made a world of difference. It is worthwhile to perform such experiments to improve the present work.
  2. Methods. The authors persevere that they froze all samples at -400° C. Please correct.
  3. Results. This part is best, because the authors here describe the obtained data and how they crunched any significant difference out of the data. However, please be a bit more critical about what you are measuring. The commercial ELISAs measure immunoreactivities. In some but not all instances pro-forms are indicated. This indicates that the authors are aware about processing events into active MMPs. Nothing, however, is written about activities, that make the biology of MMPs in OA. In this way, I disagree that this type of analysis will lead to OA biomarkers (as indicated in the abstract), because many other studies provide very similar data about rheumatoid arthritis. As long as specificity for OA is not proven, one can not write about a biomarker.
  4. Discussion. The authors mention some earlier studies (Sasaki 1994 and Yoshihara 2000) about measurements of MMPs in OA. It is essential to compare the present data with these older studies and to add the other data, e.g. from Wolfe et al. , 1993, who measured the levels of MMP-1 in synovial fluid.

Author Response

The authors would like to thank the Reviewer for all the valuable comments. The changes in the manuscript made according to the Reviewer’s suggestion were highlighted with the red bold.

Comments and Suggestions for Authors. This is an observational clinical study with the description of matrix metalloproteinase levels in a cohort of women with osteoarthritis. The association between MMPs and osteocalcin is not new. The authors have done a some literature research and incorporated this in their work. A number of issues, however, need to be addressed and these mainly relate to the limitations of this type of studies in general.  For instance, it would have been interesting to provide comparisons with the levels in other disease entities or provide links why osteocalcin was analysed aside multiple MMPs.

Major issues

  1. Introduction: it might have been interesting to provide information whether osteocalcin has been evaluated as a substrate of any MMP. In rather old studies (e.g. from the labs of Martel-Pelletier and of Tuckerman) the association between MMPs and osteocalcin was already made. A simple experiment proving that osteocalcin is a substrate of any MMP, which is found to be increased in the present and other studies, would have made a world of difference. It is worthwhile to perform such experiments to improve the present work.

The researches mentioned by the Reviewer in the term of relationship between MMPs and osteocalcin were introduced to the manuscript.

  1. The authors persevere that they froze all samples at -400° C. Please correct.

It was, of course, the typing error, the change was made according to the Reviewer’s suggestion.

  1. This part is best, because the authors here describe the obtained data and how they crunched any significant difference out of the data. However, please be a bit more critical about what you are measuring. The commercial ELISAs measure immunoreactivities. In some but not all instances pro-forms are indicated. This indicates that the authors are aware about processing events into active MMPs. Nothing, however, is written about activities, that make the biology of MMPs in OA. In this way, I disagree that this type of analysis will lead to OA biomarkers (as indicated in the abstract), because many other studies provide very similar data about rheumatoid arthritis. As long as specificity for OA is not proven, one can not write about a biomarker.

The idea of treating MMPs as a possible OA biomarkers was removed from conclusions according to the Reviewer’s suggestion. The usage of pro-forms was precluded and intended by the authors but of course it may also make up the study limitations and was added to limitation paragraph. The citation focusing on biology of MMPs was added to references.

  1. The authors mention some earlier studies (Sasaki 1994 and Yoshihara 2000) about measurements of MMPs in OA. It is essential to compare the present data with these older studies and to add the other data, e.g. from Wolfe et al. , 1993, who measured the levels of MMP-1 in synovial fluid.

The discussion paragraph was completed according to the Reviewer’s suggestion.

Reviewer 2 Report

Study is interesting, relevant for a field and potentially useful in a quest of potential biomarkers of the knee OA.

However, there are some methodological issues.  Measuring of MMP-3, MMP-9 and osteocalcin levels is common, but instead of measuring MMP-1 and MMP-13 use of precursor molecules levels, proMMP-1 and proMMP-13 is less consistent.

There are two groups of patients according to radiological Kellgren-Lawrence and Ahlback grades. It would be nice to have a third group, i.e. group of mild OA as can be seen on the knee arthroscopy procedures or a group of non obese women in case of total knee arthroplasty for advanced knee OA.

There is no comparasion of MMPs and osteocalcin levels in serum and synovial fluid according to clinical knee scores such are KSS or Oxford Knee Score.

I suggest to use the term „radiological grade“ and not „OA stage“. It should be emphasized that there are two types of the knee OA; primary (idiopathic) and secondary. In this study primary OA was discussed and authors should point it out.

Materials nad methods should be described  in more detail, especially operative technique untill time of taking the synovial fluid samples. Also, technical details should be described, for example, syringe used for taking a sample of synovial fluid and quantity of SF. Please describe  did you use tourniquet or tranexamic acid during surgical procedure because of potential interference with a results. It is unclear when the samples of peripherial blood were taken, at the time of the surgery as is stated in the abstract, or the day before the surgery, as is stated in the Materials and Methods. 

Concerning the use of corticosteriod and hyaluronans injections  as well as symptomatic slow acting drugs; patient included in the study never used such medications or just a certain period before the surgery?

Line 128 and 130: Serum was frozen to minus 400 degrees Celsius. Probably a typo.

Results

In table 2. there is a statistically significant correlation of MMP-3 serum and synovial fluid levels between radiological Ahlback I-III and IV-V grades. I do not see similar correlation of MMP-9 and MMP-13 which has been stated in the abstract and conclusion. Please explain this, maybe the explanation is in Table 3?

Please ad the number of the cases for each radiological grade in Table 3. and Table 4.

Conclusion and especially Materials and Methods are deficient, while Discussion is too extensive and hard to follow. I suggest to extend Materials and Methods as well as Conclusion, while Discussion could be a bit more concise.

Line 22: „caused by osteoarthritis“ should be „due to osteoarthritis“

Lines 42-44: „which lead to softening, ulceration, fibrillation, sclerosis of subchondral bone, loss of articular cartilage,subchondral cysts, and osteophytes“ I suggest „which lead to cartilage softening, ulceration, fibrillation and loss, sclerosis of subchondral bone as well as subchondral cysts, and osteophytes formation“.

Line 55: “pannus”. Maybe it would be better to call it “OA pannus” in order to distinguish it to RA pannus.

Line 78: citation would be nice for a statement „Osteocalcin has been thought as a serum marker of bone formation and mineralization”

Line 87: „Department of Orthopaedics and Traumatology of the Movement System” should be translated as "Department of Orthopaedics and Traumatology" or  "Department of Orthopaedics an Traumatology of Locomotor System" or "Department of Orthopaedics and Traumatology of Musculosceletal System”

Line 109: „KSS score“, please specify full name of the score and then an acronym

Lines 128 and 130: „−400 °C”   please check the actual temperature, may be a typo

Lines 127-131: and 132-136 are duplicated, please delete one of them.

Line 247: „clinical Ahlback score“. Ahlback score is radiological, not clinical score.

Line 255: „MMP-9 is reported to be the marker of obesity [43,47], in this study including obese…” - Use dot or and after brackets

Line 264: „and pathogenesis of OA [51] by causing AC degradation and pathological changes in…” I suppose, articular cartilage degeneration….

Line 271: „was1 no” – please delete1

Lines 286,287: “considered to be related to the inhibition of bone matrix mineralization [61].Dieppe et al.[62] observed that increased bone remodeling correlated with the severity of OA, and increased OC production in OA osteoblasts has been observed“ – two times word observed was used, please find another term.

Author Response

Authors would like to thank the Reviewer for all the valuable comments. The changes in the manuscript made according to the Reviewer’s suggestion were highlighted with the green bold.

Comments and Suggestions for Authors: Study is interesting, relevant for a field and potentially useful in a quest of potential biomarkers of the knee OA.

  1. However, there are some methodological issues. Measuring of MMP-3, MMP-9 and osteocalcin levels is common, but instead of measuring MMP-1 and MMP-13 use of precursor molecules levels, proMMP-1 and proMMP-13 is less consistent.

         The authors decided to use the commercial kits available at the time of experiments, that is why the pro assays were applied. Additionally, in our opinion, the information about the production on the protein level is interesting supplying with the data informing about total production not influenced by complex pathways connected with MMPs biology.

  1. There are two groups of patients according to radiological Kellgren-Lawrence and Ahlback grades. It would be nice to have a third group, i.e. group of mild OA as can be seen on the knee arthroscopy procedures or a group of non-obese women in case of total knee arthroplasty for advanced knee OA.

         The idea of obtaining the synovial fluid form knee OA on early stage was overthought by the authors. However, the amount of SF possible to be obtained before arthroscopy is insufficient for the measurements. The SF could be obtained during arthroscopy only after previous injection of BSS to the knee joint. However, such material cannot be compared to the undiluted SF obtained during TKA.

The group of non-obese women with knee OA gathered by the authors during collection of the material was too small for the statistical analysis (in whole group of TKA of 210 patients only 8 women had normal BMI). The authors continue collecting proper size samples believing that the OA in non-obese patients may constitute the distinct form of the disease.

  1. There is no comparasion of MMPs and osteocalcin levels in serum and synovial fluid according to clinical knee scores such are KSS or Oxford Knee Score.

               The comparison of MMPs and osteocalcin level were also performed for clinical and functional KSS knee scores. However, the results were similar to the results obtained using the scales evaluating the radiological grading. That is why the authors decided not to included them into the results section of the manuscript. In the table below the results of the statistical analysis are presented with values of (R) and (p) coefficients. The authors are ready to include the table to the manuscript on the Reviewer’s decision.

KSS clinical

KSS functional

pro-MMP1 serum [ng/ml]

0,063

0,6986

-0,209

0,1959

pro-MMP1 synovial fluid [ng/ml]

-0,073

0,6522

-0,183

0,2580

MMP3
serum [ng/ml]

-0,229

0,1553

-0,227

0,1588

MMP3
synovial fluid [ng/ml]

-0,224

0,1646

-0,311

0,0511

MMP9
serum [ng/ml]

0,027

0,8701

-0,276

0,0849

MMP9 synovial fluid [ng/ml]

-0,187

0,2481

-0,190

0,2402

MMP13
serum [ng/ml]

0,035

0,8286

-0,239

0,1371

MMP13
synovial fluid [ng/ml]

-0,367

0,0197

-0,252

0,1161

Osteocalcin

serum [ng/ml]

-0,399

0,0107

0,029

0,8609

Osteocalcin

synovial fluid [ng/ml]

-0,167

0,3045

0,061

0,7098

  1. I suggest to use the term „radiological grade“ and not „OA stage“. It should be emphasized that there are two types of the knee OA; primary (idiopathic) and secondary. In this study primary OA was discussed and authors should point it out.

The change was made according to the Reviewer’s suggestion.

  1. Materials and methods should be described in more detail, especially operative technique until time of taking the synovial fluid samples. Also, technical details should be described, for example, syringe used for taking a sample of synovial fluid and quantity of SF. Please describe  did you use tourniquet or tranexamic acid during surgical procedure because of potential interference with a results. It is unclear when the samples of peripherial blood were taken, at the time of the surgery as is stated in the abstract, or the day before the surgery, as is stated in the Materials and Methods.

The  following data were added to the Material and Method section according to the Reviewer’s suggestion:

The day before the surgery peripheral blood samples from all patients were obtained and centrifuged. The serum was frozen and stored at −40°C until further tests. Before the surgery started, the tourniquet was placed at the proximal side thigh. During TKA, before the incision of the knee joint capsule, the SF was obtained from the suprapatellar recess with the needle 12G, then portioned, frozen and stored at −40°C until further studies. In the case of contamination of the SF with blood, the patient was excluded from the study. Tranxeamic acid was not used during the surgery.

  1. Concerning the use of corticosteroid and hyaluronans injections as well as symptomatic slow acting drugs; patient included in the study never used such medications or just a certain period before the surgery?

The usage of the corticosteroids or/and hyaluronans injections was included as the exclusion criterium in our study (v.104), so none of the patients from our study group used ever this drugs before the surgery.

  1. Line 128 and 130: Serum was frozen to minus 400 degrees Celsius. Probably a typo.

The change was made according to the Reviewer’s suggestion.

  1. In table 2. there is a statistically significant correlation of MMP-3 serum and synovial fluid levels between radiological Ahlback I-III and IV-V grades. I do not see similar correlation of MMP-9 and MMP-13 which has been stated in the abstract and conclusion. Please explain this, maybe the explanation is in Table 3?

In Table 2 the mentioned correlations have the level of statistical tendencies for MMP-9 and MMP-13, which was described in Results section.

  1. Please ad the number of the cases for each radiological grade in Table 3 and Table 4.

The data were provided in Table 3 and 4 according to the Reviewer’s suggestion.

  1. Conclusion and especially Materials and Methods are deficient, while Discussion is too extensive and hard to follow. I suggest to extend Materials and Methods as well as Conclusion, while Discussion could be a bit more concise.

Material and Methods section was completed according to the Reviewer’s suggestion, the details are mentioned in point 5. The Conclusions were overthought and completed. The discussion paragraph was prepared for every examined MMPs and OC separately according the same scheme starting from its general properties and involvement in OA, then their concentrations and correlations with OA staging and each other. According to the Reviewer’s suggestions the authors attempt to shorten this paragraph, the removed lines are marked with  crossed out green bold

  1. Line 22: „caused by osteoarthritis“ should be „due to osteoarthritis“

The change was made according to the Reviewer’s suggestion.

  1. Lines 42-44: „which lead to softening, ulceration, fibrillation, sclerosis of subchondral bone, loss of articular cartilage, subchondral cysts, and osteophytes“ I suggest „which lead to cartilage softening, ulceration, fibrillation and loss, sclerosis of subchondral bone as well as subchondral cysts, and osteophytes formation“.

The change was made according to the Reviewer’s suggestion.

  1. Line 55: “pannus”. Maybe it would be better to call it “OA pannus” in order to distinguish it to RA pannus.

The change was made according to the Reviewer’s suggestion.

  1. Line 78: citation would be nice for a statement „Osteocalcin has been thought as a serum marker of bone formation and mineralization”

The change was made according to the Reviewer’s suggestion.

  1. Line 87: „Department of Orthopaedics and Traumatology of the Movement System” should be translated as "Department of Orthopaedics and Traumatology" or "Department of Orthopaedics an Traumatology of Locomotor System" or "Department of Orthopaedics and Traumatology of Musculosceletal System”

The change was made according to the Reviewer’s suggestion.

  1. Line 109: „KSS score“, please specify full name of the score and then an acronym

The authors decided to remove assessment according to KSS from the manuscript. Although such analysis was performed. The results did not induce any new information and conclusion but only prolong the manuscript. The sample analysis of the correlations is put as the answer on point number 3.

  1. Lines 128 and 130: „−400 °C” please check the actual temperature, may be a typo

The change was made according to the Reviewer’s suggestion.

  1. Lines 127-131: and 132-136 are duplicated, please delete one of them.

The change was made according to the Reviewer’s suggestion.

  1. Line 247: „clinical Ahlback score“. Ahlback score is radiological, not clinical score.

The change was made according to the Reviewer’s suggestion.

  1. Line 255: „MMP-9 is reported to be the marker of obesity [43,47], in this study including obese…” - Use dot or and after brackets

The change was made according to the Reviewer’s suggestion.

  1. Line 264: „and pathogenesis of OA [51] by causing AC degradation and pathological changes in…” I suppose, articular cartilage degeneration….

The change was made according to the Reviewer’s suggestion.

  1. Line 271: „was1 no” – please delete1

The change was made according to the Reviewer’s suggestion.

  1. Lines 286,287: “considered to be related to the inhibition of bone matrix mineralization [61].Dieppe et al.[62] observed that increased bone remodeling correlated with the severity of OA, and increased OC production in OA osteoblasts has been observed“ – two times word observed was used, please find another term.

The change was made according to the Reviewer’s suggestion.

Round 2

Reviewer 1 Report

I was somewhat disappointed by the minimal approach by the authors. For instance, they could have enhanced the discussion section about how their work may contribute to improve future types of analysis to develop better biomarkers, instead of just removing the sentences, which contained over-interpretations or were wrong.

Author Response

The author would like to thank the Reviewer for the remark. The authors believe that finding a biomarker of knee OA is one of  “Holy Grails” in orthopedics. Biomarkers are not only essential for the understanding of pathological pathways but also for diagnosis, prognosis and follow up [1]. With respect to the complex nature of the disease, developing new biomarkers in OA represents a major challenge, which requires strong basic research and complementary clinical and regulatory expertise. Hence, despite may years of extensive research this goal remains elusive. One of the many reasons purported as responsible for this slow pace has been the slow evolution of the understanding of the complex nature of joint tissue biology [2]. Our study showed that the mean value of MMP3 was significantly higher in patients with a more advanced disease for both serum (p = 0.0067) and synovial fluid (p = 0.0328). It allows for the hypothesis that MMP-3 level could be a diagnostic marker of knee OA completing the standard radiologic diagnostic. However, we could not prove the correlation between MMP levels in serum and SF, which diminishes the value of MMP-3 analysis. The way from the conceptualization to the proving the soluble analytes  as a biomarker is complex with identification, validation and qualification [as reviewed in 3] That is why the authors decide to resign from the conclusion about MMP-3 as possible biomarker, although the conviction of the crucial role of searching for biomarkers (both diagnostic and therapeutic [3]) in basic sciences. The authors believe that planned further researches  first with enlarged group of patients focusing on MMP-3 could highlight its role as a possible biomarker. As the authors were asked in the reviews to minimize discussion, we decided not to include this paragraph.

  1. Kraus VB, Burnett B, Coindreau J, Cottrell S, Eyre D, Gendreau M, Gardiner J, Garnero P, Hardin J, Henrotin Y, Heinegård D, Ko A, Lohmander LS, Matthews G, Menetski J, Moskowitz R, Persiani S, Poole AR, Rousseau JC, Todman M; OARSI FDA Osteoarthritis Biomarkers Working Group. Application of biomarkers in the development of drugs intended for the treatment of osteoarthritis. Osteoarthritis Cartilage. 2011 May;19(5):515-42. doi: 10.1016/j.joca.2010.08.019. Epub 2011 Mar 23. PMID: 21396468; PMCID: PMC3568396.
  2. Hunter DJ, Losina E, Guermazi A, Burstein D, Lassere MN, Kraus V. A pathway and approach to biomarker validation and qualification for osteoarthritis clinical trials. Curr Drug Targets. 2010 May;11(5):536-45. doi: 10.2174/138945010791011947. PMID: 20199395; PMCID: PMC3261486.
  3. Henrotin Y, Sanchez C, Cornet A, Van de Put J, Douette P, Gharbi M. Soluble biomarkers development in osteoarthritis: from discovery to personalized medicine. 2015;20(8):540-6. doi: 10.3109/1354750X.2015.1123363. PMID: 26954785; PMCID: PMC4819845.

Reviewer 2 Report

Thank you for your corrections, and explanations you have provided are well argumented.

Please make a few  more corrections:

MMP-1 and MMP-13 in your manuscript are pro molecules and I suggest to write them in the manuscript as: “pro-MMP1” and “pro-MMP13” in whole text, especially in the the abstract.

Please standardize the terms used for MMPs and pro MMPs through whole text and tables.

Please include the table of correlations of KSS and MMPs and osteocalcin in the manuscript, I think it makes the study more valuable.

Table 3 - pro-MMP13 instead of “pro-MMP-13”.

Table 4 – MMP-9 and MMP-3 instead of “MMP9” and “MMP3” and so on…

Line 133. proximal thigh instead of  “proximal side thigh”

Line 137.  tranexamic acid instead of “tranexexamic”

Author Response

The author would like to thank the Reviewer for the remarks. The

  1. MMP-1 and MMP-13 in your manuscript are pro molecules and I suggest to write them in the manuscript as: “pro-MMP1” and “pro-MMP13” in whole text, especially in the the abstract.

The manuscript was changed according to the reviewer suggestion.

  1. Please standardize the terms used for MMPs and pro MMPs through whole text and tables.

The manuscript was changed according to the reviewer suggestion.

  1. Please include the table of correlations of KSS and MMPs and osteocalcin in the manuscript, I think it makes the study more valuable.

The manuscript was changed according to the reviewer suggestion.

  1. Table 3 - pro-MMP13 instead of “pro-MMP-13”.

The manuscript was changed according to the reviewer suggestion.

  1. Table 4 – MMP-9 and MMP-3 instead of “MMP9” and “MMP3” and so on…

The authors standardize the terms of MMPs as: pro-MMP1, MMP3, MMP9, pro-MMP13 and use them in the hole text.

  1. Line 133. proximal thigh instead of  “proximal side thigh”

The manuscript was changed according to the reviewer suggestion.

  1. Line 137.  tranexamic acid instead of “tranexexamic”

 The manuscript was changed according to the reviewer suggestion.

This manuscript is a resubmission of an earlier submission. The following is a list of the peer review reports and author responses from that submission.